



# High-resolution measurements of elemental mercury in surface water for an improved quantitative understanding of the Baltic Sea as a source of atmospheric mercury

Joachim Kuss[1], Siegfried Krüger[2], Johann Ruickoldt[2], and Klaus-Peter Wlost[2]

[1] Department of Marine Chemistry, Leibniz Institute for Baltic Sea Research (IOW), Rostock-Warnemünde, 18119, Germany
[2] Department of Physical Oceanography and Measurement & Instrumentation, Leibniz Institute for Baltic Sea Research (IOW), Rostock-Warnemünde, 18119, Germany

*Correspondence to*: Joachim Kuss (joachim.kuss@io-warnemuende.de)

**Abstract.** Marginal seas are directly subjected to anthropogenic and natural influences from land in addition to receiving inputs from the atmosphere and open ocean. Together these lead to pronounced gradients and strong dynamic changes. However, in the case of mercury emissions from these seas, estimates often fail to adequately account for the spatial and temporal variability of the elemental mercury concentration in surface water ($Hg^0_{wat}$). In this study, a method to measure $Hg^0_{wat}$ at high resolution was devised and subsequently validated. The better-resolved $Hg^0_{wat}$ dataset, consisting of about one measurement per nautical mile, yielded insight into the sea's small-scale variability and thus improved the quantification of the sea's $Hg^0$ emissions, a major source of atmospheric mercury.

Research campaigns in the Baltic Sea were carried out between 2011 and 2015 during which $Hg^0$ both in surface water and in ambient air were measured. For the former, two types of equilibrators were used. A membrane equilibrator enabled continuous equilibration and a bottle equilibrator assured that equilibrium was reached for validation. The measurements were combined with data obtained in the Baltic Sea in 2006 from a bottle equilibrator only. The $Hg^0$ sea-air flux was newly calculated with the combined dataset based on current knowledge of the $Hg^0$ Schmidt number, Henry's law constant, and a widely used gas-exchange transfer velocity parameterization. By using a newly developed pump-CTD with increased pumping capability in the $Hg^0$ equilibrator measurements, $Hg^0_{wat}$ could also be characterized in deeper water layers. A process study carried out near the Swedish island Øland in August 2015 showed that the upwelling of $Hg^0$-depleted water contributed to $Hg^0$ emissions of the Baltic Sea. However, a delay of a few days after contact between the upwelled water and light was apparently necessary before the biotic and abiotic transformations of ionic to volatile $Hg^0$ produced a distinct sea-air $Hg^0$ concentration gradient. This study clearly showed spatial, seasonal, and interannual variability in the $Hg^0$ sea-air flux of the Baltic Sea. The average annual $Hg^0$ emission was 0.90±0.18 Mg for the Baltic Proper and to 1.73±0.32 Mg for the entire Baltic Sea, which is about half the amount entrained by atmospheric deposition. A comparison of our results with the $Hg^0$ sea-air fluxes determined in the Mediterranean Sea and in marginal seas in East Asia were to some extent similar but they partly differed in terms of the deviations in the amount and seasonality of the flux.

## 1 Introduction

In the Baltic Sea (Fig. 1), atmospheric mercury deposition decreased by about 20% between the 1990s and the start of the new millennium (Ilyin et al., 2015), but from 2006 to 2015 annual Hg deposition was relatively stable. The estimated average annual mercury deposition between 2006 and 2015 was 1.82±0.12 Mg for the Baltic Proper and 3.47±0.18 Mg for the total Baltic Sea (Gusev, 2016). Since these two values were nearly in proportion to the area, the presence of strong spatial gradients of mercury deposition was unlikely. Annual input from rivers flowing into the Baltic Sea was not well quantified because of incomplete data coverage but was reasonably estimated to be 1.1 Mg (Soerensen et al., 2016). After the transformation of mercury to its





volatile elemental form ($Hg^0$), a significant amount is emitted to the atmosphere (Kuss and Schneider, 2007;Wängberg et al., 2001).

The exchange of many gases and gaseous compounds has been studied using the film model (Broecker and Peng, 1974;Liss, 1973;Liss and Merlivat, 1986). The property of elemental mercury ($Hg^0$) that allows it to be subjected to sea-air gas exchange

is unique among heavy metals. Less water-soluble compounds, including $Hg^0$, tend to be controlled by diffusion through the water-side boundary at the sea-air interface (Jähne and Haußecker, 1998;Jähne et al., 1987). The gradient builds up on the water-side laminar layer between the bulk water and uppermost water layer. The latter is determined by the atmospheric equilibration of the gas, depending on its solubility. The gas-exchange transfer velocity has mainly been studied for carbon dioxide and is used as a standard for freshwater and seawater at 20°C (Liss and Merlivat, 1986;Nightingale et al.,

2000;Wanninkhof, 1992). Ionic mercury in seawater originates from anthropogenic emissions but also from natural sources, such as weathering and geogenic activity. Mercury enters the sea by river transport and following wet and dry deposition from the atmosphere. In surface water, it is subjected to light-dependent biotic and abiotic redox processes that transform ionic mercury to volatile $Hg^0$ (Amyot et al., 1994;Costa and Liss, 2000;Kim and Fitzgerald, 1986;Kuss et al., 2015;Mason et al., 1995), which is subjected to evasion from the sea surface. Mercury emission by the ocean accounts for about one-third of the

mercury in the atmosphere (Pirrone et al., 2013;Selin et al., 2008). However, previous calculations of marine $Hg^0$ emission fluxes were subject to several limitations; these have been partly overcome in recent years. Data coverage for emission estimates before about the year 2000 was sparse and the parameters determining $Hg^0$ flux were not well known and were thus roughly approximated (Kuss et al., 2009). Recently, the solubility of $Hg^0$ in water was newly determined (Andersson et al., 2008) and the diffusion coefficient of $Hg^0$ ($D_{Hg0}$) in water and seawater was measured (Kuss, 2014). In addition, the multi-

approach of Nightingale et al. (2000) describing the transfer velocity–wind speed relationship did gain the acceptance of many scientists in the field of gas exchange. Given the indications of an elevated transfer velocity for the Baltic Sea at higher wind speeds (Kuss et al., 2004;Weiss et al., 2007), in this study we use the parametrization of Nightingale et al. (2000).

The aims of the study were: 1) to present a new, validated method to obtain high-resolution measurements of the $Hg^0$

concentration in water ($Hg^0_{wat}$), 2) to measure the seasonal and spatial variability of $Hg^0_{wat}$ and thus of the sea-air concentration differences as a flux-driving factor, 3) to update quantification of the $Hg^0$ emission flux of the Baltic Sea based on a comprehensive dataset achieved by the new method and on recently improved knowledge of the $Hg^0$ diffusion coefficient and the solubility of $Hg^0$ , 4) to compare the magnitude and pattern of the $Hg^0$ fluxes of the Baltic Sea with those of other marginal sea areas by standardizing the fluxes according to the same transfer velocity parameterization, and 5) to newly calculate the

mercury emission budget of the Baltic Sea.

## 2 Methods

Elemental mercury in water ($Hg^0_{wat}$) and air ($Hg^0_{atm}$) was measured during research campaigns conducted between 2011 and 2015: from 24–30 July 2011 and 11–17 April 2012, onboard the r/v *Elisabeth Mann Borgese*; in 27 June–23 July 2012, onboard

the r/v *Meteor*; from 3–12 May 2013, 18–26 July 2013, and 17–29 March 2015, again onboard the r/v *Elisabeth Mann Borgese*; and from 23 July–17 August 2015, onboard the r/v *Meteor*. On almost all cruises, the Belt Sea, Arkona Sea, Bornholm Sea, and western and eastern Gotland Sea were sampled (Fig. 1) and on some cruises the bordering Åland Sea, Bothnian Sea, and the Gulf of Finland as well. A direct synthesis of the data, including the precise time and ship position, with the spectrometer output was achieved using software that enables data recording and processing and thus early inspection of the measurements.

The dataset of this study was combined with that of a study conducted in 2006 (Kuss and Schneider, 2007) for an updated flux calculation (Section 2.2).



### 2.1 Analytics

The analytics were carried out according to the trace element clean technique. The equipment was made up of carefully selected inert materials: borosilicate glass for bottles, polytetrafluoroethylene (PTFE) and polyvenylidene fluoride (PVDF) for tubing, joints, and valves. All equipment was thoroughly cleaned using a detergent (Mucasol, Merz Hygiene Co., Frankfurt, Germany), followed by diluted subboiled hydrochloric and/or nitric acids, partly under heating, and then rinsed with pure water (Milli-Q system, Millipore Co., Schwalbach, Germany).

### 2.1.1 Determinations of $Hg^0_{wat}$ and $Hg^0_{atm}$

For the atmospheric measurements, $Hg^0_{atm}$ was sampled either from air equilibrated with water or from outside air pumped from an upper deck and measured using the Tekran autonomous mercury vapor analyser (Tekran 2537A, Tekran Inc., Toronto, Canada) with a sample flow rate fixed at 1.1 L min$^{-1}$. The sampled air was guided via a gold trap within the Tekran internal dual-cartridge system to allow $Hg^0$ pre-concentration by amalgamation (Bloom and Fitzgerald, 1988;Slemr et al., 1979). The measurement cycle was divided into a 5-min sampling phase followed by analysis of the sample by desorption and peak recording. The gold trap was always cleaned prior to the next sampling interval. Since the analyzer contains two internal gold traps, continuous measurement without a gap was possible by alternate sampling (Tekran Inc., 1998). The Tekran was calibrated daily using an internal mercury permeation source, and the calibration confirmed a few times by external calibration. Beginning in 2012, $Hg^0_{wat}$ was measured using a membrane equilibrator coupled to the Tekran. The approved bottle equilibrator first implemented in 2006 (Kuss and Schneider, 2007) was then used for validation. A membrane equilibrator was already successfully applied to the measurements of dimethylsulfide in seawater (Marandino et al., 2009). In our study seawater was supplied either from the sea surface, pumped from below the ship's hull (r/v *Elisabeth Mann Borgese*) or from the bow of the ship (r/v *Meteor*) at ~3-4 m depth, or from a depth down to ~300 m, pumped by the pump-CTD. $Hg^0_{atm}$ was measured for ~1/2 h, usually once or twice a day; the measurements were then averaged and interpolated with respect to the time of the cruise, as rapid fluctuations were assumed to not significantly influence the sea-air concentration gradient.

### 2.1.2 Pump-CTD

The pump-CTD system of the Leibniz Institute for Baltic Sea Research (IOW) uses in situ sensors and water supplied from the depths of interest that is pumped to the ship's lab during profiling (Strady et al., 2008). The system consists of a standard compact CTD-rosette system (SBE 911+/SBE 32, co. SeaBird) with 11 FreeFlow water-sampling bottles (IOW with Hydro-Bios, Kiel, Germany). A positive displacement rotary vane pump (PROCON Series 3) is driven by a three-phase deep-well pump motor (Oddesse, Oschersleben, Germany). The deployment cable was newly designed to ensure good handling and stability (IOW with IG Pinnow, Germany and Falmat Cable, US). It consists of a PVDF hose of 8 mm inner diameter located in the center, 36 electrical wires, a strengthening Kevlar layer, and an outer polyurethane mantle (total diameter: 18 mm, breaking strength: 35 kN). The use of a large number of thin electrical wires keeps the total diameter of the cable small. The wires were bundled into 10 groups of 2 to 4 wires per group, with a shielding wire placed between each group, yielding shielded "twisted" pairs that could be used for the power supply of the pump motor and for optional instruments. The electrical and fluid connections between the winch drum and the CTD deck unit together with the tubing to supply the seawater were established via a specially constructed winch slip ring unit (IOW with Ramert-Hein GmbH, Kiel, Germany). The pump-CTD was used in this study to pump water from selected depths for longer time intervals at a flow rate of 4–7 L min$^{-1}$ to enable equilibrator measurements. The device is lowered or raised stepwise, followed by a temperature adaptation time of 5–10 min and a measurement interval ranging from 5 min to 1 h at each depth level. During the pump-CTD cast, the equilibrators were





permanently flushed with water supplied by the pump-CTD to keep the equilibration temperature and other parameters close to the *in situ* values. The flow rate is usually stable within a margin of error of ±0.1 L min⁻¹, with the actual flow rate monitored by a digital flow meter. Hence, the passage time of the water through the tubing is known and can be used to synchronize the data and the water flow. Additional useful features of the system are: a sonar altimeter, two laser pointers to visualize the

distance from the sea floor, and a wide-angle video camera with high-speed DSL data transmission.

### 2.1.3 Surface water pumping system

Surface water was provided by the clean seawater supply systems of the r/v *Elisabeth Mann Borgese* and r/v *Meteor* and was pumped from a depth of ~3–4 m. Seawater subjected to analyses was in contact with inert materials only and was pumped at a high flow rate to preserve as closely as possible the in situ temperature and composition. This was realized using a large-

diameter main seawater line supplying seawater at a rate of 60–70 L min⁻¹, corresponding to a current velocity of 1 m s⁻¹, in the case of the r/v *Elisabeth Mann Borgese*. Close to the equilibrators, a fraction flowing at a rate of 5–15 L min⁻¹ was diverted for measurements. The thermosalinograph was supplied by surface water from a separate pumping system to obtain the corresponding surface water salinity and temperature measurements without risk of contamination of the clean seawater supply system.

### 2.1.4 Bottle equilibrator

Seawater was taken either from the ship's clean seawater supply system or from the pump-CTD and transferred through a shower head into a 20-L equilibrator bottle (Hassa Lab, Lübeck, Germany) that allowed drainage from the bottom. Equilibrium between the supplied water and the headspace ($Hg^0_{equ}$) was achieved in about 45 min (Kuss and Schneider, 2007). Two

replicate measurements of $Hg^0_{equ}$ were usually made using the Tekran after 1 h of equilibration. From $Hg^0$ measurements of equilibrated air, the concentration in water was calculated according to Eq. (1), using Henry's law constant $H(T)$ (Andersson et al., 2008) and the equilibration water temperature ($T_{equ}$±0.02 °C) measured in the equilibrator:

$$Hg^0_{wat} = \frac{Hg^0_{equ}}{H_{T\,equ}}$$                (1)

The values of the replicate measurements were in good agreement, with a coefficient of variation of < 4%. The detection limit

of the Tekran 2537A was < 0.1 ng m⁻³.

### 2.1.4 Membrane equilibrator

The membrane equilibrator (4×13 X40 Membrane Contactor, Liqui-Cel®, Membrana GmbH, Germany) consisted of a polyethylene cylinder fed with a bundle of polypropylene membrane tubing. Seawater flowed over the outside of the hollow-

fiber membrane, around a central baffle, and back over the other side of the membrane before exiting the equilibrator (Membrana, 2016). Air sucked in by the Tekran passed first through a charcoal cartridge, then through the membrane tubing at the fixed flow rate of 1.1 L min⁻¹ and without significant resistance. Samples of equilibrated air ($Hg^0_{equ}$) were measured at 5-min intervals from the air side of the membrane equilibrator. The equilibrator continuously produced equilibrated air such that membrane equilibrator measurements could be made with the equilibrator connected directly to the Tekran. $Hg^0_{wat}$ was

also calculated following Eq. (1), with $T_{equ}$ measured directly at the equilibrator outlet. The detection limit for $Hg^0$ was < 0.1 ng m⁻³.

For most comparison measurements, the Tekran sample air intake was redirected to the bottle equilibrator headspace. Field experiments showed that the membrane equilibrator required cleaning every 3–10 days depending on the algal concentration in the water. This was recognized based on the drifting of the values compared to measurement with e.g., the bottle equilibrator


or a cleaned membrane equilibrator. The equilibrator was cleaned first by back-flushing with tap water, then with either an acid solution (100 mL subboiled $HNO_3$ (67%) filled to 2 L pure water ~3.5% v/v) or a potassium hydroxide solution (50%, 100 mL to 2 L ~ 2.5% v/v) and then a $HNO_3$ solution (5% v/v), using a gear pump (Verdergear VG 1000, Verder, Haan, Germany) for 30 min to circulate the cleaning solutions in the reversed-flow direction. The PTFE/stainless steel pump head

was connected to the membrane equilibrator by PTFE tubing. After each cleaning step the membrane equilibrator was flushed with pure water. For storage, the water side of the membrane equilibrator was dried by purging with clean air.

Beginning with the first campaign, in April 2012, the values obtained with the membrane and bottle equilibrators were in good agreement. Flow through the membrane equilibrator was varied between 1.2 and 12 L min$^{-1}$ to determine whether deviations in the equilibration value measured by the bottle equilibrator could be identified. The bottle equilibrator was operated at a flow

rate slightly higher than ~1.1 L min$^{-1}$ that was checked prior to the measurements (Kuss and Schneider, 2007). Averages obtained with the duplicate bottle equilibrator were compared with the 15-min-averages (n=3) of the membrane equilibrator shifted 15 min back to meet the approximate weighted average equilibration time of the bottle equilibrator. The agreement was good, with a deviation of ±3% only in a $Hg^0_{wat}$ concentration range of 14–38 ng m$^{-3}$ (n=53), and was confirmed many times thereafter.

**2.1.5 Total mercury ($Hg^{tot}$)**

The samples were acidified and stored until analysis at the IOW home laboratory according to the method of Bloom and Crecelius (1983). The acidified samples were subjected to permanganate oxidation and then analyzed using an automatic mercury analyzer (Mercur, Co. Analytik Jena). The sample and the tin(II) chloride reduction solution were merged in the reactor and $Hg^0$ was subsequently extracted under an argon gas stream. $Hg^0$ was then enriched on a gold/platinum net by

amalgamation, followed by desorption and analysis using CVAFS. The accuracy and precision of the method were confirmed using the BCR-579 seawater reference material (Institute for Reference Materials and Measurements, European commission, Joint Research Center) with a certified concentration of 1.9±0.5 ng $Hg^{tot}$ L$^{-1}$. The detection limit was 0.1 ng L$^{-1}$ as determined according to the calibration regression line method (DIN EN ISO 32645).

**2.2 Calculation of the Hg$^0$ sea-air exchange flux**

The film model of Liss and Slater (1974) was used to calculate the sea-air flux of gases and gaseous compounds. In that model, as shown in Eq. (2), the sea-air flux $F$ is proportional to the product of the sea-air concentration difference $\Delta C_{sea-air}$ of the gas of interest and the gas-exchange transfer velocity $k(u)$, which depends on the wind speed $u$.

$$F = k(u) \times \Delta C_{sea-air} \qquad (2)$$

Since Hg$^0$ constitutes a less soluble (Sanemasa, 1975) gaseous substance, its transfer between water and air is controlled by molecular diffusion through the water-side laminar microlayer (Jähne and Haußecker, 1998;Liss, 1983) and thus by its diffusion coefficient in water. Here we use the diffusion coefficient recently measured in freshwater and in water of oceanic

salinity (Kuss, 2014).

The gradient across the sea surface microlayer is given by the atmospheric concentration ($Hg^0_{atm}$), calculated using Eq. (1), analogous with the prevailing surface water temperature ($T_{wat}$), at the top, and the bulk water concentration ($Hg^0_{wat}$), at the bottom, of the layer. According to Eq. (2) the flux of Hg$^0$ between water and air ($F_{Hg0}$) is then obtained from both the Hg$^0$

concentration difference across the water microlayer ($\Delta Hg^0_{ml}$) and the gas-exchange transfer velocity $k(u)$, following Eq. (3):





$$F_{Hg0} = k(u) \times \left( \frac{Hg_{equ}^0}{H_{T_{equ}}} - \frac{Hg_{atm}^0}{H_{T_{wat}}} \right) = k(u) \times \Delta Hg_{ml}^0 \tag{3}$$

The transfer velocity $k$ (in cm h$^{-1}$) is mostly parameterized as a function of wind speed (u, in m s$^{-1}$) for $CO_2$ at 20°C both for freshwater ($Sc$=600) and for oceanic salinity ($Sc$=660). Here we selected the parameterization of $k$ ($Sc$=600) by Nightingale et al. (2000), shown in Eq. (4).

$$k_{600}(u) = 0.222u^2 + 0.333u \tag{4}$$

The Schmidt number $Sc$ is the ratio of the kinematic viscosity of water and the diffusion coefficient of the gas: $Sc=\nu/D$. It is used to convert $k(u)$ from $CO_2$ at standard conditions to $Hg^0$ at the prevailing temperature and salinity conditions. The temperature and salinity relationships of the diffusion coefficient and of the kinematic viscosity of water were therefore applied (Kuss, 2014). In environmental studies, a Schmidt number dependency of $k \sim Sc^{-1/2}$ is usually assumed, as shown in Eq. (5). This does not introduce a significant error because $k$ is small in the smooth surface regime below about $u$=3 m s$^{-1}$, where the dependency approaches $k \sim Sc^{-2/3}$.

$$k(u) = k_{600}(u)\left(\frac{Sc}{600}\right)^{-1/2} \tag{5}$$

The temperature dependences of $D_{Hg0}$ (in cm² s$^{-1}$) for freshwater and for seawater are given in Eqs. (6) and (7), respectively:

$$D_{Hg0}^{fresh} = 0.0335 \, e^{-\frac{18.63kJ \, mol^{-1}}{RT}} \tag{6}$$

$$D_{Hg0}^{sea} = 0.0011 \, e^{-\frac{11.06kJ \, mol^{-1}}{RT}} \tag{7}$$

where $R$ is the gas constant and $T$ the temperature in Kelvin.

For simplicity, for the Schmidt numbers given by Kuss (2014) the fitted polynomial functions dependent on $T_{wat}$ were used, as shown in Eqs. (8) and (9):

$$Sc_{S=0}^{Hg^0}(T_{wat}) = -0.0398 \, T_{wat}^3 + 3.3910 \, T_{wat}^2 - 118.02 \, T_{wat} + 1948.2 \tag{8}$$

$$Sc_{S=35}^{Hg^0}(T_{wat}) = -0.0304 \, T_{wat}^3 + 2.7457 \, T_{wat}^2 - 118.13 \, T_{wat} + 2226.2 \tag{9}$$

A weighted mean was then calculated for the brackish water of the Baltic Sea according to the measured salinity $S_{wat}$, as shown in Eq. (10):

$$Sc_{S \, wat}^{Hg^0}(T_{wat}) = \frac{1}{35}(Sc_{S=35}^{Hg^0} \times S_{wat} + Sc_{S=0}^{Hg^0} \times (35 - S_{wat})) \tag{10}$$

The comprehensive dataset comprising the $\Delta Hg_{ml}^0$ values obtained during a cruise provided a solid basis with which to calculate the fluxes for a specific season. The actual fluxes based on wind speed measured on the ship depend on the actual weather conditions, which are subject to frequent changes within the belt of westerlies of the mid-latitudes (Hagen and Feistel, 2008). For a better representation of the season, we used wind speeds based on a dataset obtained between 1951 and 2005 at



Cape Arkona (Hagen and Feistel, 2008). The flux calculations were then done by applying Eq. (3) and were based on the climatological mean and mean square wind speeds of the respective month of the year.

### 2.3 Data synthesis onboard using the *MasterTrans-Cruise assistant* software system

All hydrographic and meteorological data recorded onboard were synthesized using the IOW software system *MasterTrans-CruiseAssistant* (Wlost, IOW). The system acted as a server-based data system that combined data supplied by fixed instrumentation, permanently installed on the ship, with data generated using mobile instruments, including the mercury analyzer, brought onboard by cruise participants, to yield a comprehensive database. A standardized selection of the data and a flexible number of data from auxiliary items were then used to compile a data telegram that was transferred once per second
via the ship's local area network to all client-computers on board (Supplement, Fig. S1). Mostly these clients used the *CruiseAssistant*-Software for further processing and/or immediate visualization of the data. Most important for the organization of the sampling were the readouts from the ship's navigation system (time, position, station name, station number, course, speed, water depth), CTD/rosette system, meteorological station, and thermosalinograph. Clients could also contribute additional data to the main telegram of the *MasterTrans*-server.
For real-time data acquisition of the $Hg^0$ measurements in the framework of this study, the software *QueckIOW* (Wlost, IOW) was used. The Tekran was programmed to send the data output reports in 5-min intervals. Calibration cycles were identified by a specific label and stored separately. The Tekran output also triggered the pick-up of the equilibrator temperatures from the Kelvimat (precision electronic thermometer for two temperature probes) and of selected parameters of the *MasterTrans* main data telegram, provided by the *CruiseAssistant* running on the respective client-PC. *QueckIOW* was also used to
distinguish between the type of mercury measurements, that is, from either one of the equilibrators or the outside air. The selection of the measurement type was registered prior to the next Tekran sampling interval foreseen for the measurement change and finally included in the data table as a marker. The combination of information on the measurement type with the CTD data, from pump-CTD sampling, or from the thermosalinograph, during surface water analysis, was necessary for data calculation, validation, and interpretation. The data were visualized using screen-definable windows to display actual data
from mercury measurements and from the telegram. Preliminary data processing was achieved using the output of the temperature probe of the respective equilibrator to calculate $Hg^0_{wat}$. Using Eqs. (3)–(10), we calculated $\Delta Hg^0_{ml}$, the transfer velocity, expressed in terms of the actual wind speed, and obtained a preliminary estimate of the actual $Hg^0$ flux. The data were immediately stored line by line in an Excel-compatible spreadsheet and could thus be immediately inspected either as single values or graphs if significant changes were occurred or further action was required.

## 3 Results and Discussion

### 3.1 Surface water and atmospheric concentrations of $Hg^0$

The regional features of $Hg^0_{wat}$ during two seasons were demonstrated in two campaigns in 2013, which showed a clear spatial and seasonal variability of $Hg^0_{wat}$ (Fig. 2). Application of the 1 nm-resolution of the new membrane equilibrator measurements revealed these features at a small scale. Surprisingly strong changes in $Hg^0_{wat}$ were detected that were usually accompanied by
clear temperature and salinity changes. In May 2013, the concentration distribution of $Hg^0$ was almost uniform, ranging between 10 and 25 ng m$^{-3}$ in the Bornholm Sea as well as the western and eastern Gotland Sea. By contrast, at the northern end, in the Landsort area, it increased to ~35 ng m$^{-3}$. At the end of the cruise on May 11–12 $Hg^0_{wat}$ increased locally above 30 ng m$^{-3}$ also in the western Baltic Sea. In July, $Hg^0_{wat}$ showed a general smooth increase and an opposing trend in salinity and temperature from 54.5° to 62.5°N (Fig. 3a). This pattern was interrupted in the region between 59° and 61°N, in the northern
end of the eastern Gotland Sea and in the Åland Sea, where $Hg^0$ was strongly enriched (Figs. 2 and 3a). The $Hg^0$ concentration



increased locally from ~20 ng m$^{-3}$ to a maximum of 30−37 ng m$^{-3}$. The changes occurred in tandem with strong temperature changes (Fig. 3b). The general pattern was the same along the northward (60.0 °N; 20 July 19:30 UTC) and southward (60.1 °N; 22 July 6:00 UTC) transects, but the front shifted to the south by about ~0.1° (6 nm) within 1.5 days. The enlarged section in Fig. 3b shows the change in $Hg^0_{wat}$ of ±22 ng m$^{-3}$ that occurred in parallel with a steep temperature change of ΔT=± 2.5 K,

reflecting a sudden drop in $Hg^0_{wat}$ in the northern direction and a steep increase along the return southern route. These changes likely reflected an upwelling with a core of low temperatures and surrounded by elevated Hg$^0$ concentrations because of increased biological activity, probably of cyanobacteria, which enhanced mercury transformation. The Åland Sea archipelago was previously shown to be prone to upwelling events and to support cyanobacterial blooms during summer (Lehmann and Myrberg, 2008).

As outlined in the Methods section 2.1.1, $Hg^0_{atm}$ was measured less often, just to follow the general trend during each campaign. For July 2011 we used measurements of Hg$^0$ in air in the Arkona Sea of $Hg^0_{atm}$=1.6 ±0.4 ng m$^{-3}$ (n=7), Bornholm Sea 2.0±0.7 ng m$^{-3}$ (n=18), Eastern Gotland Sea 1.5±0.2 ng m$^{-3}$ (n=73) for the respective Hg$^0$ flux calculations. Whereas for April 2012 an average for the whole cruise of 1.19±0.12 ng m$^{-3}$ (n=22) was determined. In July 2012 we again determined averages according to the sea areas, for the Arkona Sea of $Hg^0_{atm}$=1.7 ng m$^{-3}$, Bornholm Sea 2.1 ng m$^{-3}$, Eastern Gotland Sea 1.2 ng m$^{-3}$, and

Western Gotland Sea 1.1 ng m$^{-3}$. For May 2013 a mean $Hg^0_{atm}$ for the whole cruise of 1.61 ± 0.39 ng m$^{-3}$ (n=127) and for Mar 2015 of $Hg^0_{atm}$=1.3±0.1 ng m$^{-3}$ (n=12) were calculated. For July 2015 we fitted a trend during the cruise for $Hg^0_{atm}$ of between 1.4–1.8 ng m$^{-3}$ to account for a slightly lower $Hg^0_{atm}$ in the north (mean $Hg^0_{atm}$ = 1.54 ± 0.43 ng m$^{-3}$, n=110).

**3.2 Variability of the Hg$^0$ sea-air concentration difference**

The sea-air gas exchange flux is determined by the product of $k$ and $\Delta Hg^0_{ml}$ [Eq. (3)]. The important flux-driving gradients were compiled for the different Baltic Sea areas for the respective time periods (month/year) during the campaigns (Fig. 4). In winter (Feb 2006, Mar 2015), $\Delta Hg^0_{ml}$ was usually small and its direction changeable whereas in late spring and summer (Jul 2006, Jul 2011, Jul 2012, Jul 2013) $\Delta Hg^0_{ml}$ was at times high. However, the values reflected considerable spatial differences, which were largest during spring (May 2013) but lower in autumn (Nov 2006) as expected during the transition to winter

conditions. South to north trends from the Arkona Sea to the eastern and western Gotland Sea were identified as well. These were apparently coupled to the temporal and spatial development of primary production in the Baltic Sea. The $Hg^0_{atm}$ remained relatively stable at averages of $Hg^0_{atm}$=1.0–2.1 ng m$^{-3}$, this corresponds to $Hg^0_{wat}$=4–10 ng m$^{-3}$ [Eq. (3)], unlike the Hg$^0$ concentration in surface water, which ranged from 5 to 40 ng m$^{-3}$ or even higher. The different characteristics reflected the fact that the atmosphere is relatively well mixed compared to the sea. Moreover, the transformation of ionic Hg to Hg$^0$ in

surface water is controlled by the light supply and biological production (Kuss et al., 2015) and thus by the small-scale to mesoscale variability of surface water temperature and nutrient concentrations (Lass et al., 2010).

**3.3 The contribution of coastal upwelling**

Coastal upwelling is of particular importance in summer, when a shallow thermocline restricts surface water mixing by wind,

often to just a few meters, and nutrients are depleted. Upwelling occurs frequently in the Baltic Sea (Lehmann and Myrberg, 2008), as the strong winds that predominantly blow parallel to the coast cause a shift of the surface water mass to the right in the northern hemisphere. This initiates the upwelling of colder, phosphate-enriched intermediate water to the sea surface (Lass et al., 2010), which favors the bloom of diazotrophic cyanobacteria (Wasmund et al., 2012). During the r/v *Meteor* cruise in July/Aug 2015, the ship entered an upwelling regime offshore of the island of Øland, the east coast of which is oriented west-

southwest to north-northeast (Wurl et al., 2016). Beginning on 20 July 2015, pulses of strong wind moving from the southerly to westerly direction with speeds up to 14 m s$^{-1}$ were recorded at the SMHI meteorological station Utklippan A, located near



the study site (Fig. 1). Between 28 July and 1 August 2015 the mean wind blew from 240±22° at a speed of 6.5±1.9 m s$^{-1}$, which forced an eastward Ekman offshore transport and moved intermediate water to the sea surface layer. The composition of the intermediate water differed from the replaced warm surface water, as seen in the vertical profiles of temperature, salinity, and Hg$^0$ measured at the Bornholm Deep station (Fig. 1). The temperature recording clearly showed cold winter water of about

5°C at a depth of 45–65 m (Fig. 5a). $Hg^0_{wat}$ was low (5–10 ng m$^{-3}$) in that water mass and depleted below 150 m depth under the anoxic conditions of the Fårö Deep (Fig. 5b), but $Hg^{tot}$ was also low in intermediate water (0.1–0.2 ng L$^{-1}$).

A data series from stations UPW-1 to UPW-4 was obtained during 1–4 August 2015 along a transect of ~9 nautical miles. The conditions in the upwelled water body changed during its offshore movement because of exposure to sunlight and mixing with

the surrounding warm surface water. The upwelled water at station UPW-1 had a temperature of 8.2°C, a salinity of 7.02 g kg$^{-1}$, and a Hg$^0$ concentration of ~ 11 ng m$^{-3}$ (Table 1). The salinity declined in the offshore direction to 6.82 g kg$^{-1}$ at UPW-4, whereas there was a clear increase in temperature to 15.64°C from the near-coast station UPW-1 to station UPW-4 (Table 1). The original upwelled water, with its reduced Hg$^0$ concentration and lower temperature, was assumed to be in the area of UPW-1 (Fig. 6a). At UPW-2, the increased transformation of mercury subsequently caused a clear enrichment of Hg$^0$ to 15.3

ng m$^{-3}$ in response to the elevated temperature of the surface water (Fig. 6b). At station UPW-3 the Hg$^0$ concentration in the surface water was lower, 14.3 ng m$^{-3}$ (Fig. 6c), while the value at station UPW-4 (12.2 ng m$^{-3}$) reflected a warmer mixed layer with a further reduction in the amount of Hg$^0$ (Fig. 6d), likely due to emission to the atmosphere.

The upwelled water was mostly from an intermediate level, representing the winter water from January/February of the same

year. The concentration of $Hg^{tot}$ in the cold intermediate water level ranged from 0.14 to 0.24 ng L$^{-1}$. However, transformation processes can cause a build-up of Hg$^0$ that results in a significant disequilibrium between the surface water and the atmosphere. Even at $Hg^{tot}$=0.2 ng L$^{-1}$, a transformed fraction of 10% would result in ~20 pg L$^{-1}$; with 20% subjected to transformation, the concentration would be 40 pg L$^{-1}$, corresponding to 40 ng m$^{-3}$ (see Fig. 2 for comparison). The latter would cause a strong disequilibrium compared to an average atmospheric equilibrium value of ~7.5 ng m$^{-3}$. However, an emission already started

at a mercury concentration above the atmospheric equilibrium, as ~11 ng m$^{-3}$ at UPW-1. Upwelled water affects areas as large as a few hundred square kilometers (Lass et al., 2010) and thus we conclude that upwelling contributes significantly to Hg$^0$ emissions.

### 3.4 Average seasonal mercury emissions of the Baltic Sea

The dataset of this study was analyzed together with that from a study conducted in 2006 (Kuss and Schneider, 2007) to determine an average seasonal flux pattern (Fig. 7). The calculated emission fluxes [Eq. (2)] were based on differences in the basin-average Hg$^0$ sea-air concentration measured during the cruises and on the climatological mean and mean square wind speeds (Hagen and Feistel, 2008) of the respective months for $k$ calculated using Eq. (4). The average emission fluxes ranged between a marginal uptake of 3.2 ng m$^{-2}$ d$^{-1}$ in the eastern Gotland Sea in winter and 44.8 ng m$^{-2}$ d$^{-1}$ in the eastern Gotland

Sea in July 2006. Strong emission fluxes of 35.1 ng m$^{-2}$ d$^{-1}$ in the western Gotland Sea were calculated for July 2012, 36.9 ng m$^{-2}$ d$^{-1}$ in the Arkona Sea for July 2011, and 27.3 ng m$^{-2}$ d$^{-1}$ in the Åland Sea for July 2013 (Table 2). These high Hg$^0$ sea-air fluxes were undoubtedly linked to both the light supply and seasonal primary productivity. Thus, the ΔHg$^0_{ml}$ reflected the dominating control for the flux during summer (Fig. 4). Mean wind speed and thus the transfer velocity was highest during the winter months and reached a minimum in summer (Table 2). However, in February 2006 and March 2015 high $k$ coincided

with a small sea-air concentration difference that in turn caused low fluxes of < 5 ng m$^{-2}$ d$^{-1}$. The exception was the Arkona Sea, where in March 2015 a flux of 12.6 ng m$^{-2}$ d$^{-1}$ was calculated. This occurred in tandem with a freshwater signal that might have been caused by a plume of the Odra River with elevated Hg$^0_{wat}$ and ΔHg$^0_{ml}$ (Fig. 4). The spring emission of the





western Baltic Sea was relatively high and variable. This is especially reflected in the data of the Belt Sea from May 2013 (Fig. 2), when the surface water temperatures in this shallow sea was already elevated to > 8°C instead of the 4–6°C measured in April 2006 and 2011 and in May in the other sea areas (Supplement, Table S1). The Belt Sea is prone to early spring bloom patches that may be coupled to elevated $Hg^0_{wat}$, as was the case in May 2013. In November 2006, the fluxes in all studied

regions were around 10 ng m$^{-2}$ d$^{-1}$. During autumn, the still elevated surface water $Hg^0_{wat}$ was subjected to higher wind speeds, resulting in higher $Hg^0$ fluxes compared to winter. Surprisingly, in August 2015 the fluxes were relatively low, due to the low $\Delta Hg^0_{ml}$. This likely reflected the temperatures from May to July that were slightly lower than the 30-year mean after the unusually warm winter and spring (Nausch et al., 2016). However, overall, the seasonal pattern of low fluxes in winter, high fluxes in summer, and intermediate fluxes in spring and autumn was confirmed for the Arkona Sea, Bornholm Sea, and eastern

and western Gotland Sea (Fig. 7). Nevertheless, the underlying variability and inter-annual differences were large (Table 2).

Measured $Hg^0$ sea-air concentration differences of 15–25 ng m$^{-3}$ in the eastern and western Gotland Sea in July 2006, July 2011, and July 2012 (Fig. 4) caused fluxes between 26.1 and 44.8 ng m$^{-2}$ d$^{-1}$ (Table 2). Based on a normal mixed-layer depth of 5–10 m in summer, this disequilibrium significantly declined within 1–2 weeks and was hardly restored by the reservoir in

and below the thermocline, where $Hg^0_{wat}$ showed elevated values at 20–50 m and as far down as 90 m (Fig. 5). However, $Hg^0$ production was ongoing, as supported by a $Hg^{tot}$ of ~0.1–0.3 ng L$^{-1}$, with atmospheric deposition continuing to supply Hg to surface waters.

### 3.5 $Hg^0$ emission fluxes of the Baltic Sea and other marginal seas

The average seasonal fluxes for the Baltic Sea calculated in this study were compared with those determined in previous Baltic

Sea investigations, such as by Wängberg et al. (2001), and in studies of other marginal seas, including those by Baeyens et al. (1991) and Baeyens and Leermakers (1998) in the North Sea. These investigations were followed by a study in Tokyo Bay (Narukawa et al., 2006) and by studies in the Mediterranean Sea (Andersson et al., 2007;Gårdfeldt et al., 2003) and the Baltic Sea (Kuss and Schneider, 2007), both based on equilibrator measurements with improved resolution. We also compared our results with those obtained from data from the Yellow Sea (Ci et al., 2015;Ci et al., 2011), the South China Sea (Fu et al.,

2010;Tseng et al., 2013), Minamata Bay (Marumoto and Imai, 2015), and the open East China Sea (Wang et al., 2016). However, the methods applied in the various studies differed in terms of their analytics and flux calculations (Table 3). The strongest impact on the results was most likely the differences in the *k*-parameterizations that had been applied. These included the *k(u)* of Liss and Merlivat (1986), LM86, Nightingale et al. (2000), Night2000, and Wanninkhof (1992) that was used in the form $k_{660}$=0.31u² or $k_{660}$=0.39u² (WH92) according to the type of wind speed data, as well as a parameterization determined

in the Baltic Sea (Weiss et al., 2007), Weiss2007. Since *k(u)* consists of two parts [Eq. (5)], with one being the standard *k* for Sc=600 or 660, and the other represents the term to correct for the gas under the ambient temperature and salinity conditions, one parameterization can be converted to the other by applying a factor. This was not precisely correct, especially for LM86 applied to low wind speeds (1–3 m s$^{-1}$), as a certain error is introduced. However, for wind speeds between 5 and 19 m s$^{-1}$ averaging of the respective $k_{Night2000}/k_x$ yielded relatively stable ratios with only slight trends. These ratios were

$k_{Night2000}/k_{LM86}$=1.41, $k_{Night2000}/k_{WH92-0.31}$=0.79, $k_{Night2000}/k_{WH92-0.39}$=0.62, and $k_{Night2000}/k_{Weiss2007}$=0.59 and they were subsequently used to improve the comparability of the different flux estimates (Table 3). Corresponding Night2000 values could not be estimated for the $Hg^0$ fluxes determined for the North Sea (Baeyens and Leermakers, 1998;Coquery and Cossa, 1995).

The strongest emission flux (18.3 ng m$^{-2}$ h$^{-1}$) was measured in the Yellow Sea in summer (Ci et al., 2011) and the strongest

uptake flux (−1.5 ng m$^{-2}$ h$^{-1}$) in the South China Sea in winter (Tseng et al., 2013). As estimates for the Night2000 parameterization, these values corresponded to 14.5 and -0.9 ng m$^{-2}$ h$^{-1}$, respectively. A seasonal pattern was found in most marginal sea areas (Table 3). In the Yellow Sea (Ci et al., 2011), low fluxes in winter, higher fluxes in summer, and moderate





fluxes in spring and autumn were measured, similar to the results of our study as well as those of previous studies in the Baltic Sea (Kuss and Schneider, 2007;Wängberg et al., 2001). The Mediterranean Sea (Andersson et al., 2007;Nerentorp Mastromonaco et al., 2017b), the East China Sea (Wang et al., 2016), and the South China Sea (Fu et al., 2010;Tseng et al., 2013) are characterized by high fluxes in summer and autumn. Using the Night2000, these were recalculated as 4.3 and 3.1 ng

$m^{-2}$ $h^{-1}$, 3.6 and 2.8 ng $m^{-2}$ $h^{-1}$, and 4.5/3.0 and 3.8 ng $m^{-2}$ $h^{-1}$, respectively. In Minamata Bay, an increasing trend from winter to autumn was determined: −0.9–3.8 ng $m^{-2}$ $h^{-1}$ (Marumoto and Imai, 2015). An exception to this pattern was Tokyo Bay, where the $Hg^0$ emission flux was high in winter (6.7 ng $m^{-2}$ $h^{-1}$) and lower in autumn (Narukawa et al., 2006) due to the elevated wind speeds in winter, whereas $\Delta Hg^0_{ml}$ was relatively stable during autumn/winter. In the North Sea, the fluxes were elevated in spring and autumn (4.3 ng $m^{-2}$ $h^{-1}$) but not in summer (1.9 ng $m^{-2}$ $h^{-1}$). However, the measurements were from

different North Sea areas and were made using different methods (Table 3). Hence, it is not clear whether the results were biased by the different methods and whether a lower flux in summer is representative of the North Sea, or at least was the case at that time.

### 3.6 Emission budget of the Baltic Sea

In the Baltic Proper, comprising the Arkona Sea, Bornholm Sea, and the whole Gotland Sea, average $Hg^0$ emissions were

23±50 kg in winter, 227±47 kg in spring, 435±153 kg in summer, and 211±61 kg in autumn (Table 4), corresponding to an annual emission of ~900±180 kg. For the study area, the Baltic Proper with the Belt Sea, we obtained an annual emission of 1.0±0.2 Mg $yr^{-1}$ (Table 4) that is however only ~25% of the amount estimated based on the 2006 study, which was 4.3±1.6 Mg $yr^{-1}$ (Kuss and Schneider, 2007). The difference in the two estimates was mainly attributed to two causes. First, the dominating influence of the especially high accumulation of $Hg^0$ ($\Delta Hg^0_{ml}$=27.2 ng $m^{-3}$) in the eastern Gotland Sea in summer

2006 (Fig. 4) compared to the lower (60%) average ($\Delta Hg^0_{ml}$=16.2 ng $m^{-3}$) determined during the several years of the present study. Second, the *k*-parameterization of the gas-exchange transfer velocity according to Nightingale et al. (2000) vs. that of Weiss et al. (2007) similarly resulted in a 60% lower estimate in our study than in the 2006 study. In addition, the actual wind speed measured in 2006 was higher than the climatological mean, which would account for an additional 20% reduction in the annual emission rate. Also, the present study used the Henry's law constant of Andersson et al. (2008) rather than that of

Sanemasa (1975), as was the case in the previous study. This difference reduced $\Delta Hg^0_{ml}$ by ~8% and thus the $Hg^0$ flux. This could explain the deviations of both estimates. Since the results of the sporadic measurements in other basins of the Baltic Sea were similar to those of the more frequent measurements in the Baltic Proper, the annual $Hg^0$ emission value of the latter could be extrapolated to yield an annual emission of 1.73±0.32 Mg $Hg^0$ for the whole Baltic Sea. This new annual emission estimate for the entire Baltic Sea better fits the annual atmospheric mercury deposition of 3.47±0.18 Mg (Gusev, 2016) and the annual

riverine supply of 1.1 Mg (Soerensen et al., 2016), since a large fraction is assumed to accumulate in Baltic Sea deep waters before it is finally trapped in its sediments (Kuss et al., 2017).

### 4 Conclusion

The results of studies based on high-resolution measurements better account for patchiness. In the present work, about 260 measurements of $Hg^0_{wat}$ were obtained in one day, thereby leaving some time for $Hg^0_{atm}$ determinations, calibration, and

comparison measurements. This corresponded to a spatial resolution along the transect of about one data point per nautical mile, which enabled the determination of statistically significant averages. The understanding of $Hg^0$ sea-air flux is thus improved, as the flux-driving gradient and the flux were better linked to environmental processes. However, the aim to finally constrain the sea's mercury emission is still hindered by the major source of uncertainty in calculations of mercury flux budgets; that is the method used to parameterize the relationship between transfer velocity and wind speed.




**Data availability**

Data of the study were summarized in an excel spreadsheet as average, median, standard deviation, minimum, maximum of the respective Baltic Sea areas and the respective cruises between 2006-2015 and is provided as a supplement to the main text. The original data can be obtained from J.K. and will be available in the Oceanographic Database of IOW (IOWDB) in due
time.

**Supplementary material**

Table S1: Excel spreadsheet of the mean, median, standard deviation, minimum, maximum, and number of data of salinity, temperature, $\Delta Hg^0_{ml}$, and $Hg^0$ fluxes measured on cruises of this study in the Baltic Sea between 2006 and 2015.
Fig. S1: Schematic of the data synthesis onboard r/v *Elisabeth Mann Borgese* by the *MasterTrans-Cruise assistant* software system.

**Author contributions**

J.K. Planning and organisation of mercury measurement, funding advertisement, instrumental analysis and data analysis of the
mercury measurements, manuscript writing.

S.K. Coordination of onboard instrumentation, pump-CTD development and operation with data evaluation, chief scientist, contributed to manuscript writing.

J.R. Development of the clean surface water pump system, contributed to fieldwork CTD/thermosalinograph operation and data validation, contributed to manuscript writing.

K.-P.W. Programmed software for organization of the data traffic onboard including the data telegram of the Tekran 2537A and accompanying data; CTD measurements during field campaigns, contribution to manuscript writing.

**Competing interests**

The authors declare that they have no conflict of interest.

**Acknowledgement**

We thank Stefan Weinreben, Wilhelm Kröger, Martin Kolbe, and Robert Mars for preparation and operation of the pump-CTD, *Hg$^{tot}$* Anika Ballent and Florian Cordes for support during the onboard measurements, and Hildegard Kubsch for the analysis of the samples. We also thank the chief scientists Wolfgang Roeder /S. K., Gregor Rehder, Norbert Wasmund, Ralf Prien, Volker Mohrholz, Oliver Wurl/Günther Nausch, and the captains and the crews of the r/v *Elisabeth Mann Borgese* and
r/v *Meteor.* Funding (KU 2258/2-1/2) by the German Science Foundation (DFG) is gratefully acknowledged.



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





**Table 1: Upwelling study conducted near Øland Island (Sweden) in August 2015. The station names, date and time of sampling, and the locations of the sampling sites are shown. The mean surface water temperature, salinity, and $Hg^0_{wat}$ concentration together with the corresponding standard deviation are reported.**

| Station* | Date Time (UTC) | Coordinates | Temperature (°C) | Salinity (g kg⁻¹) | $Hg^0_{wat}$ (ng m⁻³) |
|---|---|---|---|---|---|
| UPW-1 | 1 Aug 7:12 – 1 Aug 22:22 | 56.292°N,16.600°E | 8.18 ± 0.74 | 7.021 ± 0.021 | 11.1 ± 1.7 |
| UPW-2 | 2 Aug 12:52 – 3 Aug 2:02 | 56.291°N,16.662°E | 10.98 ± 0.89 | 6.952 ± 0.032 | 15.3 ± 2.8 |
| UPW-3 | 3 Aug 3:22 – 4 Aug 0:47 | 56.288°N,16.708°E | 13.37 ± 1.04 | 6.900 ± 0.031 | 14.3 ± 1.6 |
| UPW-4 | 4 Aug 3:47 – 4 Aug 22:22 | 56.291°N,16.899°E | 15.64 ± 0.31 | 6.821 ± 0.010 | 12.2 ± 0.8 |

\* Station names deviate from the original names used during the cruise.





**Table 2: Mean ± standard deviation of sea-air fluxes of elemental mercury in Baltic Sea areas based on data obtained between 2006 and 2015. Beginning in April 2012, data were obtained using a membrane equilibrator with a higher data rate. The number of data points is shown in parentheses.**

| | Feb 2006 | Apr 2006 | Jul 2006 | Nov 2006 | Jul 2011 | Apr 2012 |
|---|---|---|---|---|---|---|
| $k$ (cm h$^{-1}$) | 11.8 | 8.9 | 7.2 | 11.8 | 7.2 | 8.9 |
| | | | Hg$^0$ flux ± std (n) | | | |
| Baltic Sea area | | | (ng m$^{-2}$ d$^{-1}$) | | | |
| Belt Sea | - | 14.1 ± 2.1 (9) | 27.0 ± 5.1 (8) | 11.6 ± 4.8 (29) | - | 16.3 ± 3.8 (128) |
| Arkona Sea | 2.4 ± 1.4 (7) | 14.9 ± 5.5 (10) | 21.8 ± 3.4 (12) | 11.2 ± 3.8 (44) | 36.9 ± 1.4 (7) | 21.7 ± 8.9 (800) |
| Bornholm Sea | 0.6 ± 2.3 (12) | 8.3 ± 1.8 (94) | 24.3 ± 7.4 (8) | 6.6 ± 7.0 (27) | 29.3 ± 4.9 (18) | - |
| East. Gotl. Sea | -3.2 ± 3.0 (74) | - | 44.8 ± 6.7 (45) | 12.3 ± 4.9 (57) | 28.2 ± 4.4 (73) | - |
| West. Gotl. Sea | 1.0 ± 1.3 (39) | - | 29.9 ± 4.7 (10) | 10.4 ± 4.8 (16) | - | - |
| Åland Sea | - | - | - | - | - | - |
| Bothnian Sea | - | - | - | - | - | - |
| Gulf of Finland | - | - | - | - | - | - |

| | Jul 2012 | May 2013 | Jul 2013 | Mar 2015 | Aug 2015 |
|---|---|---|---|---|---|
| $k$ (long-term) | 7.2 | 7.8 | 7.2 | 11.2 | 7.1 |
| | | | Hg$^0$ flux ± std (n) | | |
| Baltic Sea area | | | (ng m$^{-2}$ d$^{-1}$) | | |
| Belt Sea | - | 20.3 ± 7.2 (400) | 18.0 ± 6.7 (17) | 4.6 ± 3.3 (81) | 4.2 ± 4.0 (175) |
| Arkona Sea | 20.9 ± 0.6 (7) | 14.5 ± 4.5 (472) | 16.0 ± 3.0 (144) | 12.6 ± 6.2 (317) | 7.1 ± 3.2 (348) |
| Bornholm Sea | 9.3 ± 2.4 (25) | 10.3 ± 3.1 (267) | 10.5 ± 2.3 (190) | 4.4 ± 3.5 (604) | 3.4 ± 1.6 (311) |
| East. Gotl. Sea | 26.1 ± 6.5 (1181) | 9.6 ± 3.4 (627) | 18.6 ± 7.4 (734) | 2.1 ± 2.9 (1583) | 7.7 ± 4.3 (2392) |
| West. Gotl. Sea | 35.1 ± 4.5 (496) | 17.8 ± 4.4 (376) | 12.8 ± 2.5 (65) | 2.8 ± 1.6 (104) | 7.1 ± 3.2 (1215) |
| Åland Sea | - | - | 27.3 ± 5.2 (140) | - | - |
| Bothnian Sea | - | - | 20.7 ± 1.8 (200) | - | - |
| Gulf of Finland | - | - | - | - | 1.6 ± 3.1 (251) |



**Table 3: Elemental mercury emission fluxes (in ng m⁻² h⁻¹) from selected marginal sea areas according to the season. The method used in data collection is shown as well: quantitative extraction of samples (P&T), equilibrator (Equ), or high-resolution equilibrator (Equ-high) measurements. Actual onboard measured wind speeds were used with the gas exchange model indicated: Night2000 (Nightingale et al., 2000), Weiss2007 (Weiss et al., 2007), WH92 (Wanninkhof, 1992), or other[a),b)]. All fluxes were additionally recalculated to obtain rough estimates of the fluxes according to Night2000 (see text).**

| Area/time | $Hg^0$ flux ng m⁻² h⁻¹ | Method | $k$-parameter | Night2000 ng m⁻² h⁻¹ | n | Study |
|---|---|---|---|---|---|---|
| Baltic Sea (2006–2015) | | | | | | |
| Winter | 0.1 | Equ-high/Equ | Night2000 | 0.1 | 2821 | This study [a)] |
| Spring | 0.9 | Equ-high/Equ | Night2000 | 0.9 | 3183 | This study [a)] |
| Summer | 1.6 | Equ-high/Equ | Night2000 | 1.6 | 7481 | This study [a)] |
| Autumn | 1.1 | Equ-high/Equ | Night2000 | 1.1 | 173 | This study [a)] |
| Baltic Sea (2006) | | | | | | |
| Winter | -0.1 | Equ | Weiss2007 | 0.0 | 132 | Kuss & Schneider (2007) |
| Spring | 1.4 | Equ | Weiss2007 | 0.8 | 113 | Kuss & Schneider (2007) |
| Summer | 6.7 | Equ | Weiss2007 | 4.0 | 83 | Kuss & Schneider (2007) |
| Autumn | 2.6 | Equ | Weiss2007 | 1.5 | 173 | Kuss & Schneider (2007) |
| Baltic Sea (1997–1998) | | | | | | |
| Winter | 0.8 | P&T | WH92 | 0.7 | 9 | Wängberg et al. (2001) |
| Summer | 1.6 | P&T | WH92 | 1.3 | 11 | Wängberg et al. (2001) |
| North Sea (1991–1996) | | | | | | |
| Summer | 1.9 | P&T | $k$=1 m d⁻¹ [b)] | - | 16 | Coquery & Cossa (1995) |
| Spring/Autumn | 4.3 | P&T | [c)] | - | 10 | Baeyens & Leermakers (1998) |
| Mediterranean Sea (2003–2012) | | | | | | |
| Spring | 1.5 | Equ-high | Night2000 | 1.5 | 3269 | Andersson et al. (2007) [d)] |
| Summer | 4.3 | Equ-high | Night2000 | 4.3 | ~ 3000 | Nerent. Mast. et al. (2017b) [d)] |
| Autumn | 3.1 | Equ-high | Night2000 | 3.1 | ~ 3000 | Nerent. Mast. et al. (2017b) [d)] |
| Tokyo Bay (2003–2005) | | | | | | |
| Winter | 6.7 | P&T | Night2000 | 6.7 | 18 | Narukawa et al. (2006) |
| Autumn | 4.3 | P&T | Night2000 | 4.3 | 9 | Narukawa et al. (2006) |
| Minamata Bay (2012–2013) | | | | | | |
| Winter | 1.7 | P&T | Night2000 | 1.7 | 18 | Marumoto & Imai (2015) |
| Spring | 5.3 | P&T | Night2000 | 5.3 | 18 | Marumoto & Imai (2015) |
| Summer | 4.1 | P&T | Night2000 | 4.1 | 18 | Marumoto & Imai (2015) |
| Autumn | 9.6 | P&T | Night2000 | 9.6 | 21 | Marumoto & Imai (2015) |
| Yellow Sea (2010–2012) | | | | | | |
| Spring | 1.1 | P&T | WH92 | 0.8 | 53 | Ci et al. (2015) |
| Summer | 18.3 | P&T | WH92 [e)] | 14.5 | 40 | Ci et al. (2011) |
| Autumn | 2.5 | P&T | WH92 | 2.0 | 50 | Ci et al. (2015) |
| East China Sea (2013) | | | | | | |
| Summer | 4.6 | P&T | WH92 | 3.6 | 49 | Wang et al. (2016) |
| Autumn | 3.6 | P&T | WH92 | 2.8 | 50 | Wang et al. (2016) |
| South China Sea (2003–2007) | | | | | | |
| Winter | -1.5 | FI-CVAFS [f)] | WH92 [g)] | -0.9 | 4 | Tseng et al. (2013) |
| Spring | 0.5 | FI-CVAFS [f)] | WH92 [g)] | 0.3 | 4 | Tseng et al. (2013) |
| Summer | 4.5 | P&T | Night2000 | 4.5 | 40 | Fu et al. (2010) |
| Summer | 4.8 | FI-CVAFS [f)] | WH92 [g)] | 3.0 | 4 | Tseng et al. (2013) |
| Autumn | 6.1 | FI-CVAFS [f)] | WH92 [g)] | 3.8 | 4 | Tseng et al. (2013) |

a) The flux is determined with onboard measured wind speed (Supplement Table S1).
b) A gas exchange coefficient of 1.0 m d⁻¹ is assumed (the upper estimate was used in this study).



c) A moderate wind speed of 8.1 m s$^{-1}$ was assumed and the method of Kitaigorodskii and Donelan (1984) was used.

d) As summarized by Nerentorp Mastromonaco et al. (2017a)

e) The Hg$^0$ flux was higher in summer because of episodically strong winds.

f) Combination of flow-injection with pre-concentration by gold amalgamation and cold vapor atomic fluorescence
5    spectrometric detection (Tekran 2500).

g) The method of Wanninkhof (1992) for an average wind speed (coefficient of 0.39) was used.



**Table 4: Seasonal emissions of elemental mercury (kg) in the studied areas of the Baltic Sea. The Baltic Proper comprises the Arkona Sea, Bornholm Sea, and western and eastern Gotland Sea. Extrapolation to the total Baltic Sea is based on the area ratio (see text). $Hg^0$ values are reported as the sum±standard deviation, with the number of samples shown in parentheses**

| | Area (km²) | Winter | Spring | Summer kg $Hg^0$ | Autumn | Total annual |
|---|---|---|---|---|---|---|
| Belt | 20000 | 8.4 ± 6.0 (81) | 30.8 ± 15.2 (537) | 29.8 ± 16.9 (200) | 21.2 ± 8.7 (29) | 90.2 ± 25.1 (847) |
| Arkona | 20000 | 13.6 ± 11.6 (324) | 31.0 ± 20.8 (1282) | 37.4 ± 10.4 (518) | 20.4 ± 7.0 (44) | 102.4 ± 26.9 (2168) |
| Bornholm | 40000 | 9.2 ± 15.3 (616) | 33.9 ± 13.1 (361) | 55.8 ± 34.8 (552) | 23.9 ± 25.3 (27) | 122.8 ± 47.5 (1556) |
| E Gotland | 120000 | -6.0 ± 45.7 (1657) | 105.0 ± 37.6 (627) | 273.8 ± 146.4 (4425) | 133.8 ± 53.0 (57) | 506.7 ± 166.6 (6766) |
| W Gotland | 35000 | 6.0 ± 6.5 (143) | 56.7 ± 13.9 (376) | 67.6 ± 24.5 (1786) | 33.3 ± 15.2 (16.0) | 163.5 ± 32.6 (2321) |
| Baltic Proper | 215000 | 23 ± 50 (2740) | 227 ± 47 (2646) | 435 ± 153 (7281) | 211 ± 61 (144) | 895 ± 178 (12811) |
| Study area | 235000 | 31 ± 50 (2821) | 257 ± 49 (3183) | 464 ± 154 (7481) | 233 ± 62 (173) | 986 ± 180 (13658) |
| Total Baltic | 412560 | 55 ± 88 | 452 ± 87 | 815 ± 270 | 408 ± 108 | 1730 ± 316 |




**Figures**

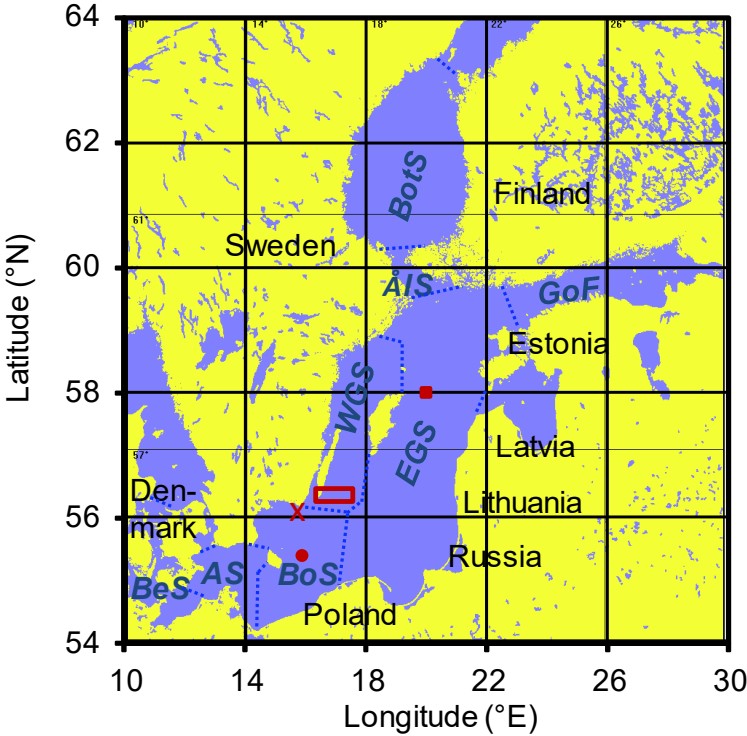

Figure 1: Map of the Baltic Sea showing the Belt Sea (BeS), Arkona Sea (AS), Bornholm Sea (BoS), Eastern Gotland Sea (EGS), Western Gotland Sea (WGS), Åland Sea (ÅlS), Bothnian Sea (BotS), and Gulf of Finland (GoF). The borders are sketched as blue dashed lines; the upwelling study site is framed in red, the SMHI's meteorological station *Utklippan A* is shown as a red cross, the Bornholm Deep station as a red solid circle, and the Fårö Deep station as a solid red square.



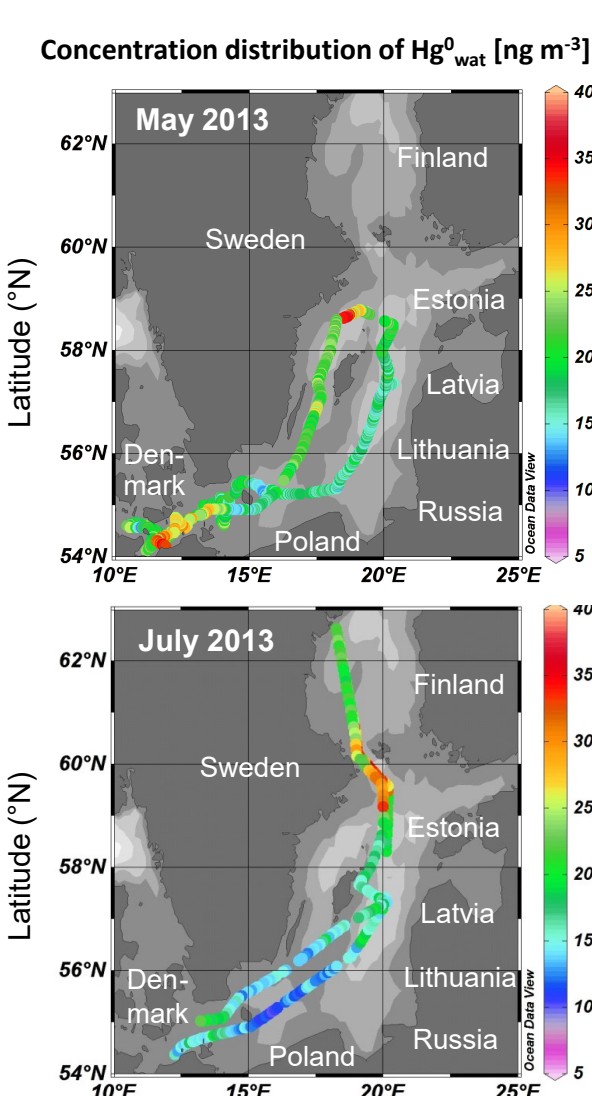

**Figure 2: The Hg⁰ surface water concentration in the Baltic Sea as determined from high-resolution *Hg⁰wat* measurements conducted in May 2013 (upper panel) and July 2013 (lower panel) was plotted by using *Ocean Data View* (Schlitzer, 2014).**



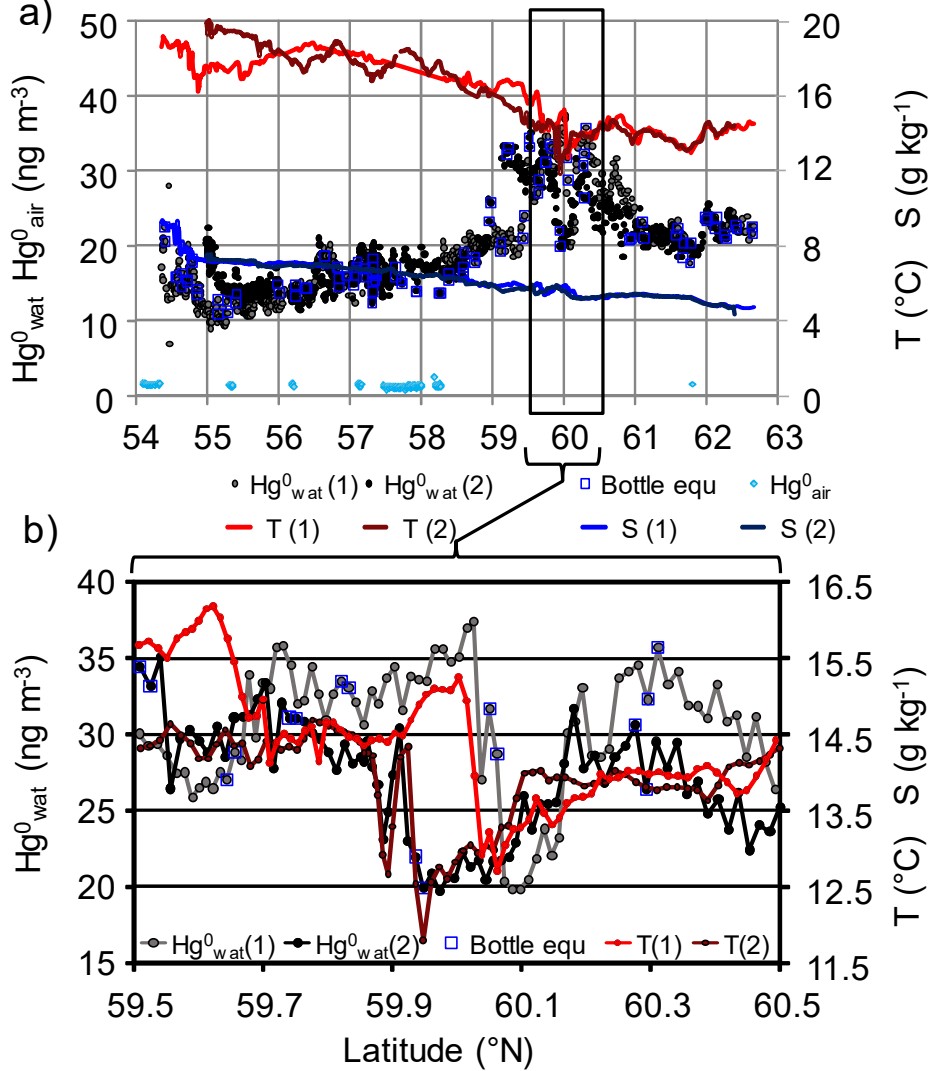

**Figure 3: Hg⁰ concentration, temperature (T), and salinity (S) data obtained from surface water measurements in July 2013 along a northward (grey, red, blue) and southward (black, dark red, dark blue) transect, respectively. (a) Data for the whole campaign and (b)** *Hg⁰_wat* **and temperature in the Åland Sea area only. Bottle equilibrator measurements are indicated by blue open squares. Hg⁰ atmospheric measurements are given as light blue diamonds in (a) for comparison.**



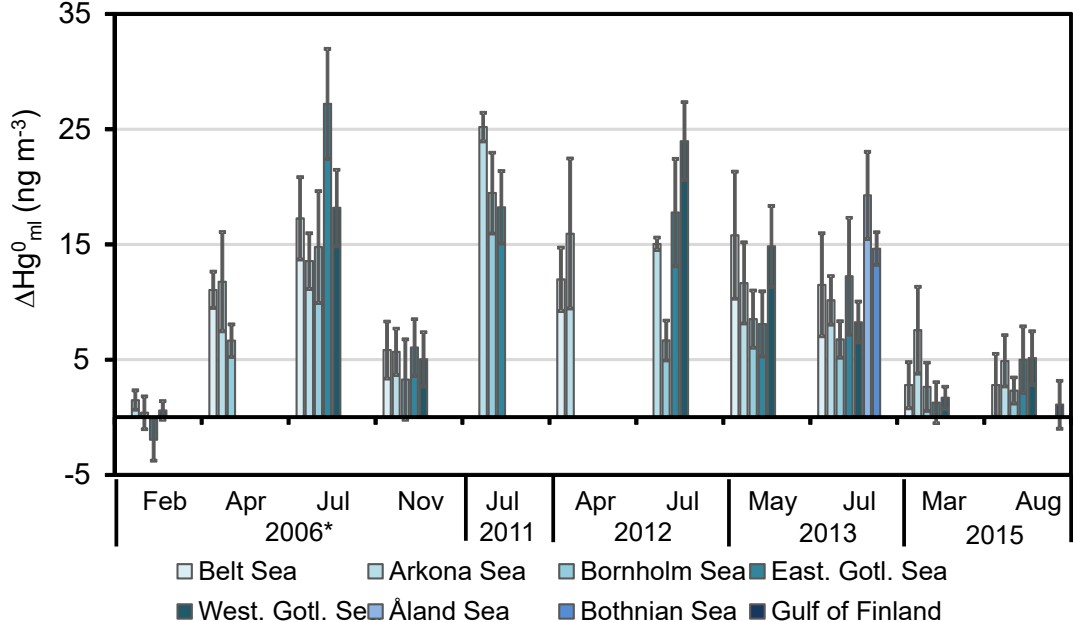

**Figure 4: The Hg⁰ sea-air concentration difference that determined the flux is given for the different sea areas (see map in Fig. 1) during the cruises of this study (2011–2015). * For comparison, data from a 2006 study (Kuss and Schneider, 2007) are provided.**





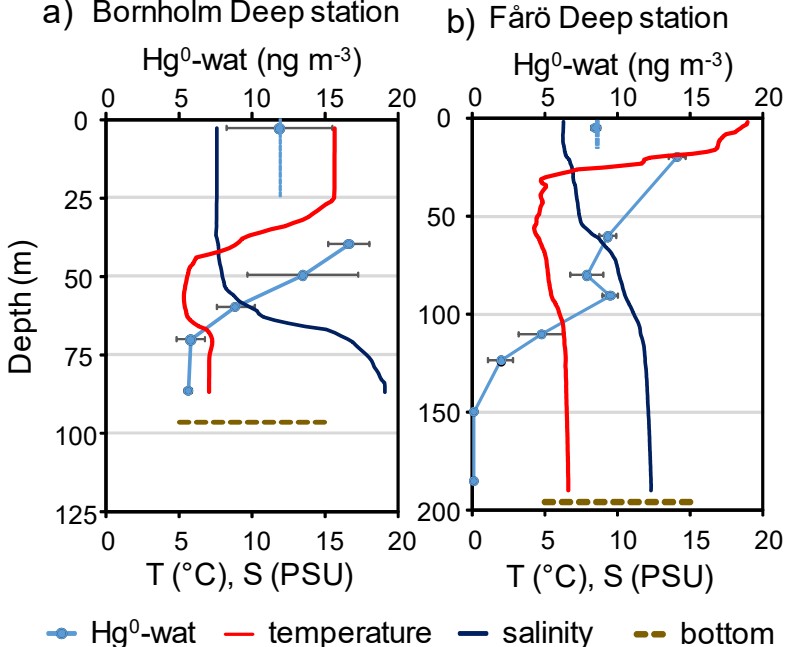

Figure 5: The vertical profiles of Hg$^0$, temperature (T), and salinity (S) at the (a) Bornholm Deep and (b) Fårö Deep stations.



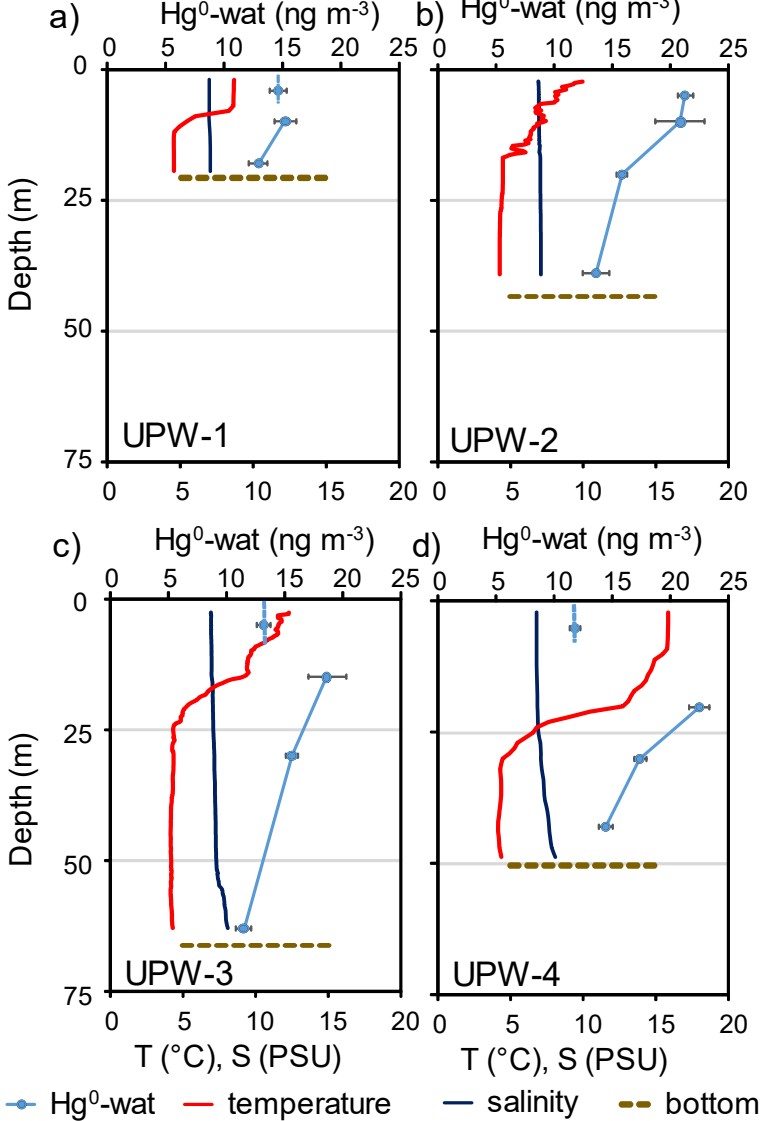

**Figure 6: The vertical profiles of $Hg^0_{wat}$, temperature (T), and salinity (S) of an upwelling area close to Øland Island (Sweden) measured in 2015. The four stations UPW-1, UPW-2, UPW-3, and UPW-4 were separated by two, one, and six nautical miles, respectively.**





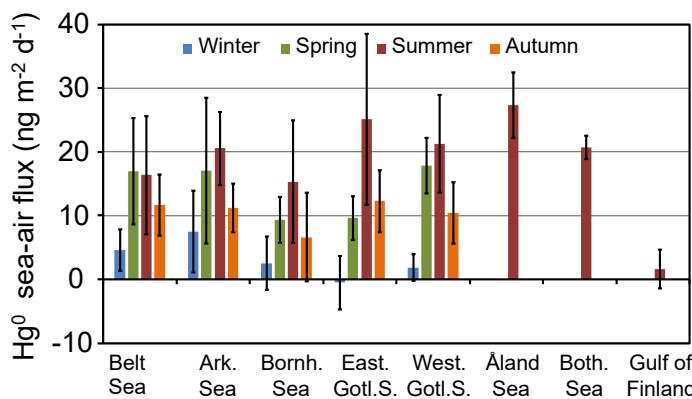

**Figure 7: The average seasonal mercury emission of selected Baltic Sea areas determined based on the data from this and a previous study (Kuss and Schneider, 2007). Standard deviation is indicated. In the Gulf of Finland, only a small area was sampled.**

