# Peer review of "High-resolution measurements of elemental mercury in surface water for an improved quantitative understanding of the Baltic Sea as a source of atmospheric mercury"

_Atmospheric Chemistry and Physics, 2017_

## Referee Comment (RC1) · Anonymous Referee #1 · 16 Nov 2017

16 November 2017

Anonymous review of Kuss et al., 2017, High-resolution measurements of elemental mercury in surface water for an improved quantitative understanding of the Baltic Sea as a source of atmospheric mercury, *ACPD*

**General comments**

Kuss and collaborators present high-resolution measurements of $Hg^0$ in seawater and air-sea fluxes in the Baltic Sea. High-resolution measurements make an important scientific contribution to the field. Ocean emission are large global source of Hg to the atmosphere and, as pointed out by the authors, there is considerable uncertainty in air-sea fluxes and so I'm very glad to see the authors working in this area.

I recommend the manuscript of publication after revisions. The manuscript is clearly written and logically organized. The greatest area for improvement is Section 3. Section 3 currently reads as a dense report-out on results and is a bit too light on the discussion. It would greatly improve the manuscript to add more insight and context to Section (i.e., tell the reader why the results matter, how the results change or add to existing knowledge, and the implications).

**Specific comments**

Page 1, line 16: The use of "major" is ambiguous. Major in what context? A major source in the Baltic region? In the global context, it's small (<1% of global ocean emissions). Consider deleting "major", replacing with a quantitative statement, or clarifying the context in which it's a major source.

Page 1, lines 18-23: "A membrane equilibrator enabled continuous… $Hg^0_{wat}$ could also be characterized in deeper water layers." This level of details seems more appropriate the Methods section than the Abstract.

Page 8, lines 10-17: This paragraph is especially dense with numbers. Consider summarizing in a table instead of the main text.

Page 9, lines 25-27: "Upwelled water affects areas…. We conclude that upwelling contributes significantly to $Hg^0$ emissions." This seems like an important result and merits further elaboration. Why does this matter? How does it change or add to the current understanding of what's going on in the Baltic or other marginal seas?

Page 11, lines 21-22: A 60% difference is substantial. If Nightingale 2000 and Weiss 2007 yield such different results, what's the implication for current global budgets of ocean emissions?

Section 3.6: What the relationship between the emission budget presented for the Baltic Sea and the trends stated in the introduction (decline since 1990s, relatively flat since 2006)?

Data availability: I strongly encourage the authors to make the un-averaged data available, in addition to the averaged data. Un-averaged data will be of greatest interest to modelers want to compare simulated and measured values.

Figure 3: It's really hard to distinguish the symbols for $Hg^0_{wat}(1)$ and $Hg^0_{wat}(2)$. I'd suggest using two colors with greater contrast.

---

## Referee Comment (RC2) · Anonymous Referee #2 · 19 Jan 2018

Air-sea exchange is one of the major uncertainties in understanding the global mercury cycle. The presented study improves on previous work by Kuss et al. in the Baltic Sea. Based on high resolution measurements it gives novel insights into important processes and short term variability of air-sea exchange. This is a valuable contribution to research on mercury.

Moreover, the paper is well written, the methodology robust and the measurements trustworthy. I support publication of the manuscript after a few issues are addressed:

**Major points:**

**Page 7 line 1-2: Please discuss the error introduced by the usage of average wind speeds as compared to high resolution (e.g. hourly) data.**
**As wind speed is squared ($0.222\ u^2 + 0.333u$) even using the median instead of the mean might have a large impact on the calculated fluxes.**
**I think you need at least estimate the error due to the averaging. Especially in early autumn when $Hg^0$ concentrations are still high and storms are more common.**

**Page 11 line 26-31: This section needs to be clarified:**
**The 1.73Mg $Hg^0$ annual evasion is supposed to be the estimate for the whole Baltic Sea? So the Bothnian Sea, Bay of Bothnia, Bay of Finland, Bay of Riga have a combined $Hg^0$ flux of only 730 kg?**

**How do you extrapolate the data to get to this conclusion? This leads to many unanswered questions and I would ask you to give more information on the extrapolation method and its uncertainty (e.g. Do you consider the effects of sea ice? Do you have measurement data for the Bay of Riga and Bay of Finland? What is the effect of average wind speeds?)**

Minor points:

Page 1 line: I suggest that you also cite the HELCOM reports (2007 & 2011) on which the Soerenson et al. riverine influx estimate is based on.

Page 2 line 24: "The aims are" instead of "The aim were"
This clarifies that you are talking about the actual study and not a previous one.

Page 5 line 13: Please clarify: "of ±3% only in a $Hg^0_{wat}$ concentration range of 14–38 ng m$^{-3}$"
To me that means that the 3% error was only validated for concentrations in the range of 14-38ng/m³. If this is the case the question arises how large the error is outside this range. Otherwise I suggest to drop the word "only" which makes the sentence clearer.

Page 8: lines 5-6: This finding is based on average wind speeds. How well does this capture storm events?

Page 9, lines 25-27: This is a great result. It would be interesting if you could estimate the impact of an upwelling event on the mercury flux that would normally occur without this event. This could also be a source for inter-annual variability due to shifts in wind fields.

Page 11 lines 21-22: You identify a 60% difference in calculated air-sea flux due to differences in parametrizations. How important do you thing the inter-annual variability is in comparison. And how large is the effect of averaging wind speeds in comparison?

Page 11 line 26-31: I suggest that you compare the results of your extrapolation with modelling results which can be seen as a more sophisticated way of extrapolating measurement data (e.g. Soerenson et al., 2016; Bieser and Schrum et al., 2016).

Page 11 line 37: fluxes instead of flux.

I agree with reviewer #1:
*"Data availability: I strongly encourage the authors to make the un-averaged data available, in addition to the averaged data. Un-averaged data will be of greatest interest to modelers who want to compare simulated and measured values."*

---

## Author Comment (AC1) · 20 Feb 2018

Responses to reviewer #1 comments and changes made according to the suggestions

General comments

Kuss and collaborators present high-resolution measurements of Hg0 in seawater and air-sea fluxes in the Baltic Sea. High-resolution measurements make an important scientific contribution to the field. Ocean emission are large global source of Hg to the atmosphere and, as pointed out by the authors, there is considerable uncertainty in air-sea fluxes and so I'm very glad to see the authors working in this area.

I recommend the manuscript of publication after revisions. The manuscript is clearly written and logically organized. The greatest area for improvement is Section 3. Section 3 currently reads as a dense report-out on results and is a bit too light on the discussion. It would greatly improve the manuscript to add more insight and context to Section (i.e., tell the reader why the results matter, how the results change or add to existing knowledge, and the implications).

> In Section 3 we present a follow up from the "surface water and atmospheric concentrations of Hg0" to the resulting flux driving gradients "Variability of the Hg0 sea-air concentration difference", considering local peculiarities by discussing "the contribution of coastal upwelling". Then we provide an estimate of the emitted amounts of mercury by calculating "the average seasonal mercury emissions of the Baltic Sea" by using climatological wind speed data. Actual fluxes during our campaigns are subsequently compared to the observations in other marginal sea areas ("Hg0 emission fluxes of the Baltic Sea and other marginal seas"). Thereby we emphasize the different methods in measurement and calculation. Finally, we provide an "emission budget of the Baltic Sea". We discuss the impact of different k-parameterizations as well as of other controlling parameters and quantify the emission according to current knowledge. It is clear to report things that are supported by data and it was not the aim to go too far beyond the determined Hg0 emission fluxes. However, we explained some observations in more detail in the revised manuscript: Upwelling (Page 9, Lines 23-28), atmospheric contribution (Page 10, Lines 16-18).

Specific comments

Page 1, line 16: The use of "major" is ambiguous. Major in what context? A major source in the Baltic region? In the global context, it's small (<1% of global ocean emissions). Consider deleting "major", replacing with a quantitative statement, or clarifying the context in which it's a major source.

> It was aimed as a general remark, however it appeared misleading. We modified the sentence to distinguish between local findings and the global meaning (Page 1, Line 16).

Page 1, lines 18-23: "A membrane equilibrator enabled continuous… Hg0 wat could also be characterized in deeper water layers." This level of details seems more appropriate the Methods section than the Abstract.

> ➢ It is aimed as a brief summary of the applied methods. Since the methods reflect an important part in the paper, it appears adequate to give this summary in the abstract. No changes were made (Page 1, Lines 18-23).

Page 8, lines 10-17: This paragraph is especially dense with numbers. Consider summarizing in a table instead of the main text.

> ➢ We summarized the data of fitted and averaged atmospheric $Hg^0$ measurements in a table, now Table 1. A brief introduction to Table 1 is given (Page 8, Lines 10-15).

Page 9, lines 25-27: "Upwelled water affects areas…. We conclude that upwelling contributes significantly to $Hg^0$ emissions." This seems like an important result and merits further elaboration. Why does this matter? How does it change or add to the current understanding of what's going on in the Baltic or other marginal seas?

> ➢ The coastal upwelling itself is a challenging subject. Its spatial and temporal variability make it difficult - even by using sophisticated modelling - to quantify the contribution, so we decided to further explain the phenomenon and to include a reference that shows episodic upwelling areas in the Baltic Sea (Lehmann and Myrberg, 2008) (Page 9, Lines 23-28).

Page 11, lines 21-22: A 60% difference is substantial. If Nightingale 2000 and Weiss 2007 yield such different results, what's the implication for current global budgets of ocean emissions?

> ➢ It is really a serious problem for sea-air flux calculations. However, as discussed in the paper, the Nightingale approach appeared a suitable compromise. The also often used parameterization of Wanninkhof (Wanninkhof, 1992) was later re-calculated by using an extended data base (Sweeney et al., 2007). It revealed a "new Wanninkhof" which was clearly close to the Nightingale parameterization. Hence, there is some confidence that the relationship between k and u is constrained somewhere around Nightingale's parameterization. Thus, the uncertainty is expected to be reduced for most environmental conditions by using it. No changes were made (Page 11, Lines 20-22).

Section 3.6: What the relationship between the emission budget presented for the Baltic Sea and the trends stated in the introduction (decline since 1990s, relatively flat since 2006)?

> ➢ Based on Figure 4 by looking at the summer month data, a slight declining trend might be deduced in the sea-air Hg0 concentration difference. However, the variability is large and the Baltic Sea hydrography is complicated by frequent upwelling and inflow events in time intervals of several years. Thus considering spatial variability and almost decadal changes a trend is not trustworthy unless the consequences of the different processes on $Hg^0$ emission are basically understood – which seems currently not the case. Therefore, we didn't feel ready to make conclusion on this point. (No changes were made).

Data availability: I strongly encourage the authors to make the un-averaged data available, in addition to the averaged data. Un-averaged data will be of greatest interest to modelers want to compare simulated and measured values.

> ➢ The data are now available on request from the IOW data base (details are given on Page 12, Lines 4-5).

Figure 3: It's really hard to distinguish the symbols for $Hg^0_{wat}$ (1) and $Hg^0_{wat}$ (2). I'd suggest using two colors with greater contrast.

> ➢ We modified the symbols of $Hg^0_{wat}$ (1) to a lighter grey to better distinguish both data sets.

**References used for the answers**

Lehmann, A., and Myrberg, K.: Upwelling in the Baltic Sea — A review, J. Mar. Syst., 74, S3-S12, 10.1016/j.jmarsys.2008.02.010, 2008.

Sweeney, C., Gloor, E., Jacobson, A. R., Key, R. M., McKinley, G., Sarmiento, J. L., and Wanninkhof, R.: Constraining global air-sea gas exchange for $CO_2$ with recent bomb $^{14}C$ measurements, Global Biogeochem. Cycles, 21, GB2015, doi:10.1029/2006GB002784, 2007.

Wanninkhof, R.: Relationship between wind speed and gas exchange over the ocean, J. Geophys. Res., 97, 7373-7382, 10.1029/92JC00188, 1992.

---

## Author Comment (AC2) · 20 Feb 2018

Responses to reviewer #2 comments and changes made according to the suggestions

Air-sea exchange is one of the major uncertainties in understanding the global mercury cycle. The presented study improves on previous work by Kuss et al. in the Baltic Sea. Based on high resolution measurements it gives novel insights into important processes and short term variability of air-sea exchange. This is a valuable contribution to research on mercury. Moreover, the paper is well written, the methodology robust and the measurements trustworthy.

I support publication of the manuscript after a few issues are addressed:

Major points:

Page 7 line 1-2: Please discuss the error introduced by the usage of average wind speeds as compared to high resolution (e.g. hourly) data. As wind speed is squared ($0.222 u^2 + 0.333u$) even using the median instead of the mean might have a large impact on the calculated fluxes. I think you need at least estimate the error due to the averaging. Especially in early autumn when Hg0 concentrations are still high and storms are more common.

> As we pointed out in the respective sentence (now it is Page 6 Line 36 to Page 7 Line 1&2), we used mean wind speeds and mean square wind speeds. This reliably avoids the averaging bias. The mean square wind speed accounts for the original wind speed variance. Thus, it appears a feasible method to use climatological data sets without the averaging bias, if mean square wind speeds are available (no changes were made).

Page 11 line 26-31: This section needs to be clarified: The 1.73 Mg Hg0 annual evasion is supposed to be the estimate for the whole Baltic Sea? So the Bothnian Sea, Bay of Bothnia, Bay of Finland, Bay of Riga have a combined Hg0 flux of only 730 kg? How do you extrapolate the data to get to this conclusion? This leads to many unanswered questions and I would ask you to give more information on the extrapolation method and its uncertainty (e.g. Do you consider the effects of sea ice? Do you have measurement data for the Bay of Riga and Bay of Finland? What is the effect of average wind speeds?)

> We deduced 1.73 Mg for the whole Baltic Sea based on the areas given in Table 4 (now Table 5). The whole Baltic Sea represents an area of 412 560 km², the study area is 235 000 km² representing ~57% of the whole Baltic Sea. The extrapolation was based on the area ratio and was argued as likely realistic because of the sporadic measurements that were done in the Bothnian Sea and the Bay of Finland. Winter was not considered as an important season of emission, hence ice coverage was not explicitly mentioned. Potential accumulation of $Hg^0$ below ice would likely be released after cracking and melting of the ice coverage in spring. We don't think that this would change the estimate significantly.
> We added the term "according to the area ratio" (Page 11, Lines 27-28)

Minor points:

Page 1 line: I suggest that you also cite the HELCOM reports (2007 & 2011) on which the Soerenson et al. riverine influx estimate is based on.

> Yes, we included the "5th Baltic Sea pollution load compilation" as the important reference behind (Page 2, Line 1).

Page 2 line 24: "The aims are" instead of "The aim were" This clarifies that you are talking about the actual study and not a previous one.

➢ We made the change accordingly (Page 2, Line 24).

Page 5 line 13: Please clarify: "of ±3% only in a Hg0wat concentration range of 14–38 ng m−3" To me that means that the 3% error was only validated for concentrations in the range of 14-38ng/m³. If this is the case the question arises how large the error is outside this range. Otherwise I suggest to drop the word "only" which makes the sentence clearer.

➢ That's right. "only" was introduced to emphasize that the deviation is small. However, it is obviously misleading and we deleted the word "only" (Page 5, Line 12-14).

Page 8: lines 5-6: This finding is based on average wind speeds. How well does this capture storm events?

➢ We couldn't figure out to which number this is referred to, however, as commented on the first major point, we used mean and mean square winds that omits an averaging bias, i.e., it accounts for the distribution between very low and high wind speeds (No changes were made).

Page 9, lines 25-27: This is a great result. It would be interesting if you could estimate the impact of an upwelling event on the mercury flux that would normally occur without this event. This could also be a source for inter-annual variability due to shifts in wind fields.

➢ Upwelling is a complex spatial and temporal process {Lehmann, 2008 #2840}. We give some more information about upwelling in the revised version (Page 9, Lines 23-28). Unfortunately, based on our data set a detailed quantification of the impact on emission is not possible, it would have required a complete spatial and temporal coverage of the upwelling area and the adjacent area by measurements, which was not possible during our campaign.
Yes indeed, shifts in the wind fields certainly contribute to the inter-annual variability of the Baltic Sea Hg0 emission. However, other meteorological parameters are important as well (solar radiation, cloudiness …).

Page 11 lines 21-22: You identify a 60% difference in calculated air-sea flux due to differences in parametrizations. How important do you thing the inter-annual variability is in comparison. And how large is the effect of averaging wind speeds in comparison?

➢ The inter-annual variability is basically reflected in Figure 7, where seasonal averages have been calculated. It appears to be about 30-50% in spring and summer. We think that the parameterization of Nightingale et al. is a good choice, probably less than 25% uncertainty. As commented on the first major point, we used mean and mean square winds that omits an averaging bias, i.e., it accounts for the distribution between very low and high wind speeds (No changes were made).

Page 11 line 26-31: I suggest that you compare the results of your extrapolation with modelling results which can be seen as a more sophisticated way of extrapolating measurement data (e.g. Soerenson et al., 2016; Bieser and Schrum et al., 2016).

➢ Yes indeed modelling appears the only way to account for the variable influences on the Hg0 gas exchange with a full area coverage. However, we are a bit cautious on this suggestion, as on our point of view modelling did not and could not account for all drivers operating on mercury emissions by the sea in a proper way. Current state of the art requires more measurements and process studies that would enable an improved modelling in the future (No changes were made).

Page 11 line 37: fluxes instead of flux. I agree with reviewer #1: "Data availability: I strongly encourage the authors to make the un-averaged data available, in addition to the averaged data. Un-averaged data will be of greatest interest to modelers who want to compare simulated and measured values."

➢ We exchanged flux by fluxes (Page 11, Line 37).
➢ The data are now available on request from the IOW data base (details are given on Page 12, Lines 4-5).